# DNA Methylation at a Single Locus of Human Genome Accurately Recapitulates Episignature of CREBBP-Related Rubinstein–Taybi Syndrome [note 1]

**DOI:** 10.3390/ijms26189183

**Published:** 2025-09-19

**Authors:** Olga A. Zemlianaia, Alexey I. Kalinkin, Alexander S. Tanas, Anna V. Efremova, Ilya V. Volodin, Olga R. Ismagilova, Anton S. Smirnov, Dmitry V. Zaletaev, Marina V. Nemtsova, Sergey I. Kutsev, Vladimir V. Strelnikov

**Affiliations:** 1Research Centre for Medical Genetics, 115522 Moscow, Russia; o.zemlianaia@gmail.com (O.A.Z.); alexeika2@yandex.ru (A.I.K.); tanas80@gmail.com (A.S.T.); anv.efremova@gmail.com (A.V.E.); ilya.fevral@mail.ru (I.V.V.); ismolga.mg@gmail.com (O.R.I.); anton.smirnov.9910@gmail.com (A.S.S.); zalnem@mail.ru (D.V.Z.); nemtsova_m_v@mail.ru (M.V.N.); kutsev@mail.ru (S.I.K.); 2Department of Bioinformatics, Biomedicine Institute, Pirogov Russian National Research Medical University, 117513 Moscow, Russia; 3Department of Translational Medicine and Biotechnology, Sechenov University, 119991 Moscow, Russia

**Keywords:** chromatinopathies, episignature, DNA methylation, Rubinstein-Taybi syndrome

## Abstract

The disruption of the epigenetic mechanisms of gene expression regulation due to the emergence of pathogenic variants in genes-encoding elements of epigenetic machinery leads to the development of chromatinopathies. This group of hereditary diseases includes 179 syndromes, some of which present with overlapping phenotypes. Despite the variety of approaches to molecular diagnostics of chromatinopathies, it is not always possible to establish the molecular diagnosis by traditional methods; thus, the issue of optimizing diagnostic algorithms remains relevant. One of the most rapidly expanding areas of post-genomic molecular diagnostics is episignature detection, which relies on genome-wide DNA methylation analysis. This article aims to represent an original approach to indirect diagnostics of chromatinopathies on the example of Rubinstein–Taybi syndrome 1, which is based on the analysis of the methylation level of a limited set of loci designed to reproduce its classic episignature. In the current study, we apply two methods of targeted quantitative analysis of DNA methylation, which are relatively accessible and can be integrated into diagnostic practice. We demonstrate that Rubinstein–Taybi syndrome 1 episignature may be successfully reduced to a single locus of human genome, and that quantitative bisulfite DNA methylation analysis at this locus allows accurate identification of the Rubinstein–Taybi syndrome 1 patients.

## 1. Introduction

Epigenetic regulation of gene expression is one of the most significant factors ensuring adequate development of tissues and organs. Disruption of epigenetic mechanisms caused by pathogenic variants in the genes participating in the epigenetic machinery can lead to neurodevelopmental diseases, also referred to as chromatinopathies [1,2]. These syndromic disorders share certain clinical manifestations including intellectual disability, growth abnormalities and congenital anomalies [3,4].

Rubinstein–Taybi syndrome (RTS) is an autosomal dominant disorder associated with pathogenic variants in the *CREBBP* (RTS1; OMIM: 180849) or *EP300* (RTS2; OMIM: 613684) genes. The products of these genes, CBP and p300 proteins, respectively, possess histone acetyltransferase activity, thus defining Rubinstein–Taybi syndrome as a chromatinopathy. The main clinical features include intellectual disability, facial dysmorphism (downslanted palpebral fissures, convex nasal ridge with low-hanging columella, high palate, grimacing smile, talon cusps), abnormalities of the extremities, growth deficiency and microcephaly [5,6]. Differential diagnosis includes other hereditary syndromes such as Wiedemann–Steiner syndrome (OMIM: 605130), Bohring–Opitz syndrome (OMIM: 605039), genitopatellar syndrome (OMIM: 606170), Cornelia de Lange syndrome (CDLS1, OMIM: 122470; CDLS2, OMIM: 300590), Kabuki syndrome (Kabuki syndrome 1, OMIM: 147920; Kabuki syndrome 2, OMIM: 300867), Say–Barber–Biesecker–Young–Simpson syndrome (OMIM: 603736) and Floating–Harbor syndrome (OMIM: 136140) [5,7]. Molecular genetic testing for RTS involves several options; however, a significant number of patients presenting with distinctive phenotypes remain undiagnosed [8,9,10,11]. For instance, molecular testing of *CREBBP* and *EP300* in one of the largest clinical RTS cohorts consisting of 395 probands managed to establish the diagnosis in only 37% of the cases [11].

One of the possible approaches to chromatinopathies diagnostics optimization involves identification of a disorder-specific genome methylation pattern, also known as episignature. DNA methylation signatures have been extensively studied for the past decade and proved themselves to be useful in both distinguishing the epigenomes of patients and healthy people and genetic variants interpretation [3,12,13,14,15]. Usually, episignature detection is carried out by genome-wide methods such as methylationEPIC BeadChip array, which can be challenging for some laboratories to integrate into diagnostic practice. In this regard, it seems reasonable to study the possibility of using a confined set of specific regions of the genome with most significant methylation level changes as a biomarker of a particular chromatinopathy.

In this paper, we demonstrate the way of performing episignature detection with accessible targeted methods based on bisulfite deep-targeted NGS and on Sanger bisulfite sequencing of specific loci on the example of Rubinstein–Taybi syndrome. Although Sanger bisulfite sequencing has always been considered a method suitable only for qualitative assessment of DNA methylation, technology has been developed that allows us to perform a quantitative analysis as well [16]. Targeted sequencing methods are much less labor-intensive and expensive compared to wide-genome DNA methylation analysis methods, making them attractive tools for the identification of the aberrantly methylated loci.

In order to form a sufficient sample set, given the higher prevalence of variants in the *CREBBP* gene compared to variants in the *EP300* gene, we decided to focus only on RTS1 for this study. Moreover, RTS1 episignature is intriguing to study in terms of limitations of applicability of targeted methods for measuring DNA methylation levels, since the maximum difference in the methylation level between individual CpG dinucleotides discriminating RTS1 patients and healthy controls is rather faint, not exceeding 20–25% [12,13,17]. An interesting task is not only to detect such low values of difference in DNA methylation level by targeted methods but also to study the sensitivity of RTS1 episignature itself, which was previously determined to be insufficient for use in diagnostic practice [18].

## 2. Results

### 2.1. Selection of a Limited Set of Loci Potentially Recapitulating RTS1 Episignature

The most representative changes in terms of CpG dinucleotides methylation level in patients with RTS1 compared to healthy controls (Table 1) were selected from the study by Y. Tang et al. [17]. The selection of CpG dinucleotides and further target loci compilation were conducted following the three criteria: (1) the difference in methylation level of the selected locus between RTS1 and control samples should equal or exceed 20%; (2) one target locus should contain at least two CpG dinucleotides; (3) mutual arrangement of CpG dinucleotides should allow PCR design with inserts not exceeding 300 bp, suitable for sequencing.

The RTS1 episignature [17] that we used in our study to select a limited set of loci is based on data obtained from Illumina Infinium methylationEPIC BeadChips (850k). EPIC BeadChips were designed to assess methylation at selected scattered CpG dinucleotides but not in their continuous sequences. Our criteria of amplicon design listed above imply analysis of immediate neighboring CpGs; thus, CpGs flanking those with EPIC BeadChips Probe IDs and not included in the existing RTS1 episignature were mechanistically included in our design. As a consequence, the following sites were added to the study that are absent from the EPIC BeadChips design:-chr6:10555905 at locus 1.1;-chr6:10556052, chr6:10556069, chr6:10556098, chr6:10556174, chr6:10556199, chr6:10556204, chr6:10556244 at locus 1.2;-chr2:206628415, chr2:206628484, chr2:206628491, chr2:206628521, chr2:206628525, chr2:206628529, chr2:206628531, chr2:206628539 at locus 2.1;-chr2:206628592, chr2:206628606, chr2:206628609, chr2:206628621, chr2:206628645 and chr2:206628714 at locus 2.2.

In order to level out possible sequencing signal fluctuations, nearby CpG sites with similar methylation behavior were considered as one feature based on their mean methylation level. According to this principle, sites with genomic coordinates chr6:10556204 and chr6:10556199, chr6:10556107 and chr6:10556147, chr2:206628521 and chr2:206628525, chr2:206628529 and chr2:206628531, chr2:206628645 and chr2:206628692, and chr2:206628727 and chr2:206628737 were united, though during machine learning these CpGs were considered separately as well.

In order to amplify and further sequence the loci of interest, we have designed pairs of bisulfite PCR primers; see Section 4 for details.

### 2.2. DNA Methylation Quantitative Analysis

Loci of interest were successfully amplified from bisulfite treated templates and further sequenced using both targeted NGS and Sanger methods.

Methylation level information (Appendix A) was successfully obtained and extracted from NGS data. The average sequencing coverage of the studied loci was 8000X.

On the other hand, Sanger bisulfite sequencing failed to assess the methylation level of several CpGs (Appendix A). The cytosines with genomic coordinates chr6:10555905, chr6:10556174, chr6:10556244, chr2:206628415, chr2:206628592, chr2:206628606 and chr2:206628609 were not assessed by Sanger sequencing at all, and cytosines with genomic coordinates chr6:10555808, chr6:10556052, chr6:10556107, chr6:10556147, chr6:10556199 and chr2:206628553 were assessed in a limited number of samples. Occasional loss in methylation level data were observed in cytosines with genomic coordinates chr6:10555881, chr6:10556204, chr6:10556174, chr2:206628525, chr2:206628529, chr2:206628531, chr2:206628539, chr2:206628625, chr2:206628645 and chr2:206628747.

### 2.3. Feature Selection

Feature selection based on high-throughput sequencing (NGS) data established the methylation level of the chr6:10556199_chr6:10556204 site as the most significant predictor for the sample binary classification (Figure 1). The calculation of the main parameters of the diagnostic test showed that the most accurate classification model is based on the methylation level of chr6:10556199_chr6:10556204 site alone (Appendix A).

As for Sanger bisulfite sequencing, the locus that demonstrated the greatest importance in terms of predictive ability was the one that included cytosines with genomic coordinates chr6:10556107 and chr6:10556147 (Figure 2). However, the highest values of sensitivity, specificity and accuracy were demonstrated by the classification model based on two loci, chr6:10556199_chr6:10556204 and chr6:10556147_chr6:10556107. Single-predictor models have a number of advantages over multiple-predictor models including simplified sample preparation process and the ability to obtain results based on the cut-off value alone, with no need of using a model every single time. Thus, we continued the study choosing the same locus as a predictor as in the case of high-throughput sequencing, since among the single-predictor models the one based on chr6:10556199_chr6:10556204 methylation level performed the best (Appendix A).

### 2.4. Target Locus DNA Methylation RTS1/Control Cut-Off

Figure 3 shows the result of the control and RTS1 samples distribution into two classes depending on the methylation level of the chr6:10556199_chr6:10556204 site measured by high-throughput bisulfite sequencing. The cut-off value of the corrected methylation level was 0.5690, which corresponds to a methylation level of 84.45%.

As for Sanger bisulfite sequencing, the result of the distribution of normal and pathological samples into two classes depending on the methylation level of the chr6:10556199_chr6:10556204 locus is illustrated in Figure 4. In this case, the cut-off value of the corrected methylation level calculated using the support vector machine on the training set samples was 0.581809, which corresponds to a methylation level of 80.99%.

### 2.5. Samples with the CREBBP Gene VUS and from Phenotypically Overlapping Syndromes

To assess overall classification ability of our model, we combined the training and test set samples, as well as added samples with variants of uncertain significance (VUS) in the CREBBP gene and samples from patients with syndromes phenotypically overlapping with RTS1 to the analysis (Figure 5).

The same operation was performed with a classification model based on Sanger bisulfite sequencing (Figure 6).

### 2.6. Sensitivity, Specificity and Accuracy of the Classification Models

ROC analysis demonstrates that both assembled classification models perform with high values of sensitivity, specificity and accuracy (Figure 7). Ten-fold cross-validation was applied towards the training set but not the test set. The ROC curves show that the classification model based on Sanger bisulfite sequencing generally outperforms the classification model based on targeted high-throughput bisulfite sequencing in predicting the attribution of a sample to the RTS1 or control class.

## 3. Discussion

In the current study, we demonstrate that the analysis of the methylation level at only one locus of the genome can classify samples as RTS1 or non-RTS1 with high accuracy. The RTS1 genome-wide episignature was successfully derived in several recent studies [12,13,17], and it was shown to be highly specific. We demonstrate how the reduction in the number of studied CpGs from hundreds to just two of them does not result in the loss of accuracy. Both targeted high-throughput bisulfite sequencing and Sanger bisulfite sequencing showed impressive results in capturing minor differences in DNA methylation level between the control and RTS1 samples. These approaches have numerous advantages over genome-wide techniques, including rapidity and simplicity of execution and data processing, and can be easily implemented into routine diagnostic practice as tools for improving RTS1 detection.

The data obtained in the current study also provides basis for further research concerning the functional significance of the aberrantly methylated chr6:10556199_chr6:10556204 locus in RTS1 patients. This site is located in the promoter region of the *GCNT2* gene, which encodes the enzyme responsible for formation of the blood group I antigen. It could be valuable to investigate if hypomethylation of the chr6:10556199_chr6:10556204 locus affects the expression of *GCNT2*. It is not clear yet whether the abnormal level of expression of this gene could contribute to the RTS1 phenotype or not; however, the existing data from ChIP-seq database suggests that p300 is recruited to the area in which the studied CpGs are encompassed, particularly in neural cells.

ROC analysis indicates that the model based on Sanger bisulfite sequencing predicts Rubinstein–Taybi syndrome 1 more accurately compared to the model based on targeted high-throughput bisulfite sequencing. However, some loss in DNA methylation level data are observed while using Sanger bisulfite sequencing compared to targeted high-throughput bisulfite sequencing. The lack of data corresponding to certain cytosines in several samples could have occurred due to the imperfection of the context-based signal level measurement, which is inevitable because of the variability of the electropherograms. This also might have been the reason for the discrepancy between the outcome of the feature selection and the predictor that the most accurate classification model was based on. As the methylation level of cytosines with genomic coordinates chr6:10556147, chr6:10556107 and chr6:10556199 were not assessed in certain samples, the decision on the significance of the chr6:10556107_chr6:10556147 along with chr6:10556199_chr6:10556204 loci as a predictor system was made by the algorithm based on incomplete data. On the contrary, all of the studied CpGs were equally well sequenced in all of the samples using targeted high-throughput bisulfite sequencing. Due to the adequacy of the data, the NGS method confirmed that the methylation level of the chr6:10556199_chr6:10556204 locus is the most representative and significant predictor. ROC analysis established a model based on Sanger bisulfite sequencing as the more accurate one, but a more complete DNA methylation dataset suggests that the targeted high-throughput bisulfite sequencing-based model is preferable.

Despite demonstrating high sensitivity, specificity and accuracy, both classification models still produced a number of mistakes, incorrectly classifying five samples. It is clear that the pathogenicity of the variants detected in the misclassified samples cannot be questioned based on the current study result. The methylation level values of the chr6:10556199_chr6:10556204 locus for each of those samples are relatively close to the threshold value and form a “borderline” zone that determines the measurement error. In order to establish it more accurately, it is necessary to continue the study by increasing the sample size.

Our research also shows that the methylation level of the chr6:10556199_chr6:10556204 locus in patients with syndromes phenotypically overlapping with RTS1 differs from that in patients with RTS1. Interestingly, two samples from patients with Wiedemann–Steiner syndrome and Floating–Harbor syndrome were initially referred for molecular diagnostics of *CREBBP* and *EP300* genes because the patients demonstrated clinical features consistent with RTS; however, pathogenic variants in the *KMT2A* and *SRCAP* genes, respectively, were found instead within further exome sequencing. This finding indicates the fact that chr6:10556199_chr6:10556204 methylation level reflects a disruption in function of the *CREBBP* gene regardless of phenotype. Nevertheless, more confident use of episignature analysis for differential diagnosis requires analyzing a larger number of DNA samples from patients with phenotypically overlapping syndromes.

It meets our expectations that samples with variants of uncertain significance in the *CREBBP* gene were assigned to the RTS1 class. Episignature identification is a promising tool for interpreting the clinical significance of genetic variants, and the possibility of using episignature analysis for diagnosis confirmation has been studied by experts [19,20,21,22]. It was previously suggested that a specific methylation pattern could be considered as a characteristic phenotype (PP4 criterion according to ACMG-AMP classification) along with the PM level of strength, slightly increasing the likelihood of pathogenicity of the variant of uncertain significance [19,23]. In one of the most recent studies regarding the use of episignatures in diagnostic practice the authors propose a list of recommendations for the episignature testing results interpretation [24]. It is suggested that the presence of specific episignature along with a suspected causative variant or a VUS in the gene associated with the identified episignature allows us to apply either PS3 or PP4 criterion with varying levels of strength [24,25]. The PS3 criterion stands for a proven damaging effect of the particular variant on a gene product and is usually based on a reliable functional study [23]. Specific DNA methylation profile might be considered a functional effect of gene disruption; however, the mechanistic relationship between the specific episignature and a particular genetic variant cannot always be confirmed. There was also a proposal to introduce an additional criterion related to the episignature analysis, namely, a strong criterion PS5: “A DNA methylation signature matching the gene in which the variant was identified is present” [26]. Overall, high accuracy of the episignature analysis is expected to be reflected in the weight of the corresponding criterion, but its use to determine pathogenicity or benignity of a particular variant calls for caution. The absence of a characteristic episignature does not necessarily rule out the associated syndrome due to the yet unexplored heterogeneity of epigenotype, and a positive episignature testing result does not always indicate the unconditional pathogenicity of the detected variant [27].

Overall, despite the fact that both models showed remarkable results in classifying samples from healthy individuals and patients with RTS1, Wiedemann–Steiner syndrome, Floating–Harbor syndrome and Cornelia de Lange syndrome, it is crucial to note the limitations of the current study. The existing state of the Sanger bisulfite sequencing methodology does not allow us to measure the DNA methylation level as thoroughly as in the case of high-throughput bisulfite sequencing, which makes it difficult to draw any conclusions regarding the comparison between the performances of the models. It should be noted that there is a slight difference in the methylation levels obtained by these two methods in our study; thus, it is crucial to avoid mixing the data up while assembling the machine learning-based classification model. Furthermore, it is necessary to test the reduced RTS1 episignature on larger cohorts of patients with VUS in the *CREBBP* gene and patients with phenotypically overlapping disorders in order to better understand its diagnostic value and discriminating accuracy.

The demonstrated approach to reducing the minimum number of CpG dinucleotides tested for episignature detection represents an effective way of diagnostics optimization and can be applied to many other inherited syndromes with known episignatures. Furthermore, detection of the most stable elements (loci) of the episignature would provide better understanding of pathogenesis of the disease and potential insights into therapeutic approaches.

## 4. Materials and Methods

### 4.1. Study Cohorts

DNA samples were derived from patients referred for genetic testing after their phenotype was evaluated by a clinical geneticist. Control DNA samples were derived from healthy individuals with no specific chromatinopathy phenotype. In this study, a total amount of 99 samples was analyzed, including (1) 60 DNA samples from patients with established pathogenic/likely pathogenic variants in the *CREBBP* gene, (2) 30 DNA samples from healthy controls, (3) 5 DNA samples from patients with detected variants of uncertain significance in the *CREBBP* gene, (4) 4 DNA samples from patients with syndromes phenotypically overlapping with Rubinstein–Taybi syndrome, Wiedemann-Steiner syndrome (1 sample), Floating-Harbor syndrome (1 sample) and Cornelia de Lange syndrome (2 samples) (Appendix A).

### 4.2. Design of a Limited Set of Loci Potentially Recapitulating RTS1 Signature

The most representative changes in terms of methylation level (difference in methylation level between RTS1 and control samples equal or exceeding 20%) in patients with RTS1 compared to healthy control CpG dinucleotides (Table 1) were selected from the study by Y. Tang et al. [17]. Primer pairs for bisulfite PCR (Table 2) were designed to quantitatively assess DNA methylation at the selected loci, following two criteria: (1) one target locus should contain at least two CpG dinucleotides; (2) mutual arrangement of CpG dinucleotides should allow PCR design with inserts not exceeding 300 bp, suitable for sequencing.

### 4.3. Sample Processing

DNA was extracted from peripheral blood leukocytes using standard techniques. Bisulfite conversion was performed with the EpiTect Fast Bisulfite Kit (QIAGEN, Hilden, Germany), following the manufacturer’s protocol. The libraries for the massive parallel sequencing were prepared according to the manufacturer’s protocol of the Prep&Seq™ U-target DNA basic module (Parseq Lab, Saint Petersburg, Russia). High-throughput sequencing was performed on NextSeq500 (Illumina, San Diego, CA, USA). Sample targeted high-throughput bisulfite sequences for a patient with RTS1 and for a healthy control are presented on Appendix A, and on Appendix A.

Sanger bisulfite sequencing was performed in accordance with the technique that allowed us to assess DNA methylation level quantitatively. In order to create a scale for measuring the methylation level at target points, considering the dependence of the amplitude of each electropherogram signal on the nucleotide context, model amplicons that contain only nonmethylated cytosines were produced by amplifying target regions of a DNA sample followed by bisulfite conversion. The experimental samples were subjected to bisulfite conversion, after which the target region was amplified and sequenced. Sanger sequencing was performed on a 3500 Genetic Analyzer (Applied Biosystems, Waltham, MA, USA). Sample Sanger bisulfite sequences for a patient with RTS1 and for a healthy control are presented on Appendix A, and on Appendix A.

### 4.4. Sequence Data Processing and Methylation Level Calculation

High-throughput sequencing data processing included the following: (1) quality control before and after adapter trimming using FastQC v. 0.12.1 [28]; (2) trimming adapters, primers, homopolymers, low-quality read ends and length filtering (maximum length 260 nucleotides, minimum length 100 nucleotides) using fastp v. 0.23.4 program [29]; (3) alignment to the sodium bisulfite-converted genome using Bismark v. 0.25.1 and bowtie2 v. 2.5.4 software [30,31]; (4) filtering reads to a panel using Samtools v. 1.18 program [32]; (5) extraction of methylation data using bismark_methylation_extractor v. 0.25.1 software; (6) sorting and indexing of bam-format files; (7) searching for sequence variants to detect cytosine to thymine substitutions using BSsnper v. 1.1 software [33]; (8) splitting of bam-file into separate locus-dependent files. Methylation level data were obtained by comparing the abundance of cytosine and thymine at CpG dinucleotide target sites.

Electropherograms obtained from Sanger sequencing were processed by R script. To calculate the methylation level of each studied CpG site, the median was first calculated for each corrected signal in model amplicons with nonmethylated cytosines only. A percentage scale was used to assess methylation. In order to bring the corrected methylation level values under this scale, the 100% methylation level was divided by the median in each individual signal peak and, thus, the scaling coefficient scale was calculated. Then, to obtain the difference in methylation level, the value obtained in the previous step was multiplied by the signal level of each nucleotide and subtracted from 100% (Appendix A). The R script is available upon request.

### 4.5. Machine Learning

In order to identify CpGs with methylation level which would allow dividing the studied samples into two classes in the best possible way, feature selection was performed using the random forest method with Boruta R package version 6.0.0 [34]. To establish the most profitable combination of features, the training set samples were used to calculate sensitivity, specificity and accuracy of the classification models based on every possible combination of features that were considered important, as well as features separately.

To create a binary classification model, the caret R package version 1.7–2 was used to train a support vector machine (SVM) [35]. The dataset for SVM was formed with control (N = 30) and RTS1 samples with pathogenic/likely pathogenic variants in the *CREBBP* gene (N = 60). It was randomly split between train and test sets in a ratio 70:30, respectively. Also, a 10-fold cross-validation was performed for a model performance estimation. The values of sensitivity, specificity and accuracy of each classification model were obtained using receiver operating characteristic (ROC) analysis conducted with caret R package [35].

The results of machine learning were presented with a linear decision rule using a cut-off value to distribute the studied samples of the training and test sets into classes, “control samples” (non-RTS1) and “RTS1 samples”. Corrected methylation level was obtained using Z-transformation. The cut-off value was calculated based on the model parameters.

## Figures and Tables

**Figure 1 ijms-26-09183-f001:**
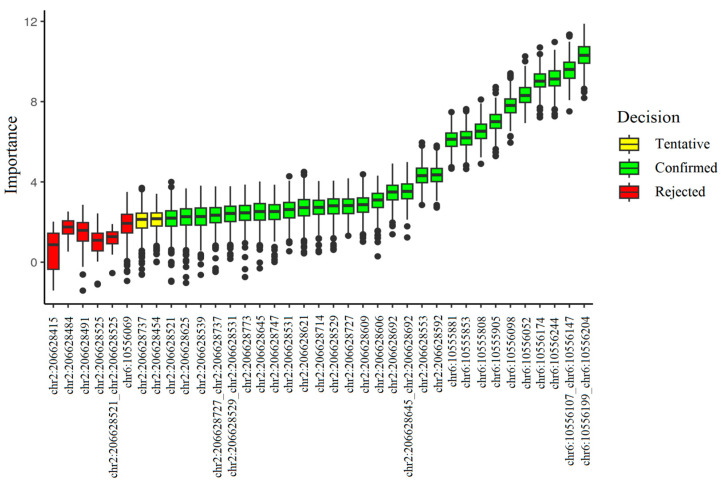
The result of feature selection based on targeted high-throughput bisulfite sequencing data. Methylation levels of 29 CpG sites out of 37 were interpreted as important predictors; for 2 sites the decision was tentative and 6 features were determined to be unimportant.

**Figure 2 ijms-26-09183-f002:**
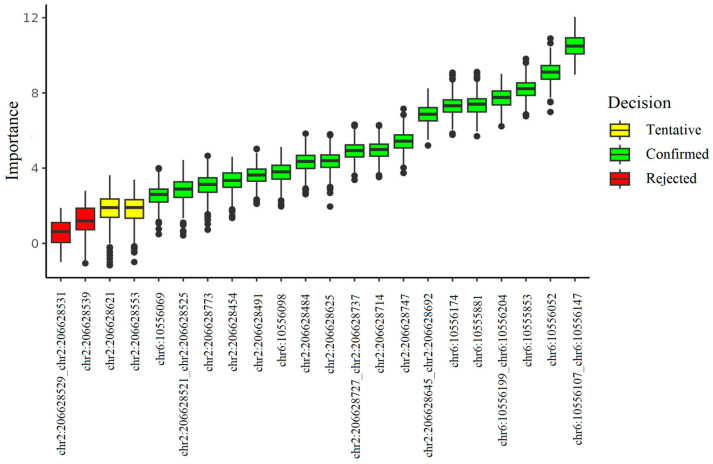
The result of feature selection based on Sanger bisulfite sequencing data. Methylation levels of 18 CpG sites out of 22 were interpreted as important predictors; for 2 sites the decision was tentative and 6 features were determined to be unimportant.

**Figure 3 ijms-26-09183-f003:**
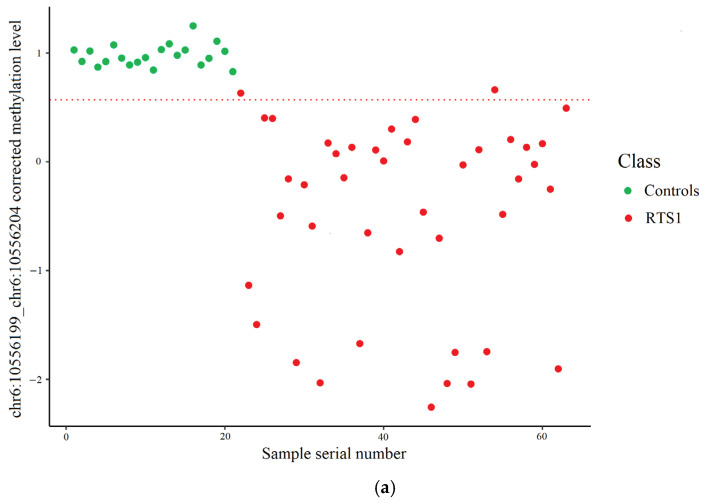
Control and RTS1 samples distribution using targeted high-throughput bisulfite sequencing data. The dashed line represents the cut-off value. (**a**) Training set: two RTS1 samples incorrectly classified. (**b**) Test set: one control and two RTS1 samples incorrectly classified.

**Figure 4 ijms-26-09183-f004:**
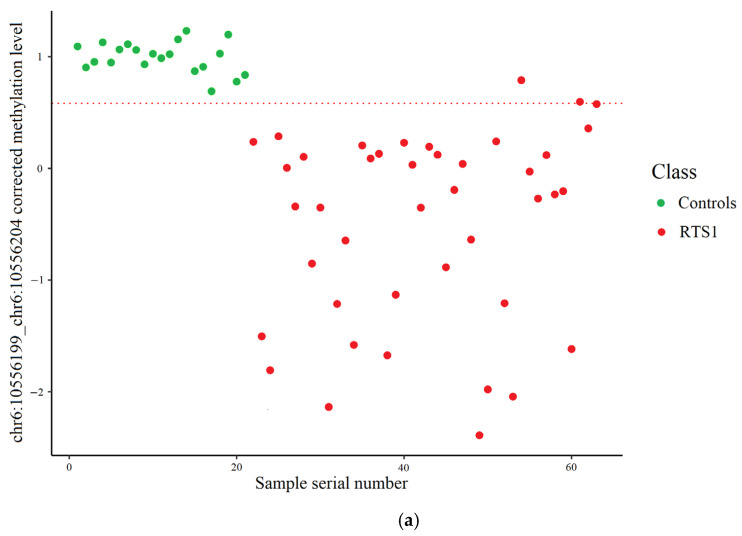
Control and RTS1 samples distribution using Sanger bisulfite sequencing data. The dashed line represents the cut-off value. (**a**). Training set: three RTS1 samples incorrectly classified. (**b**). Test set: one control and one RTS1 sample incorrectly classified.

**Figure 5 ijms-26-09183-f005:**
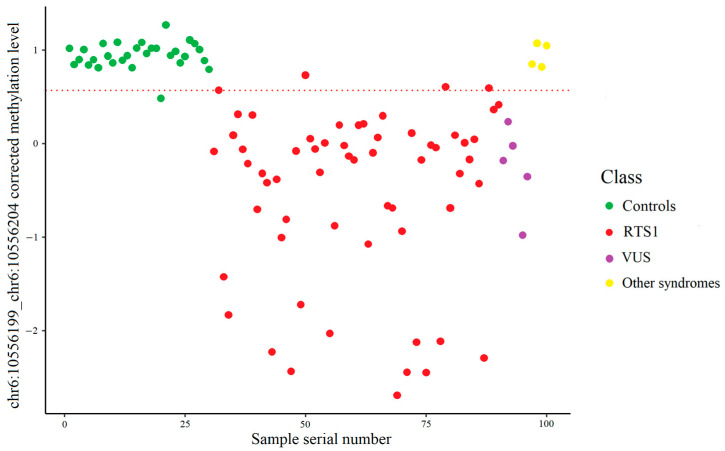
Full dataset samples classification by model based on targeted high-throughput bisulfite sequencing. All of the samples with VUS in the *CREBBP* gene were classified as RTS1. All of the samples from patients with phenotypically overlapping syndromes clustered above the cut-off line were classified as controls.

**Figure 6 ijms-26-09183-f006:**
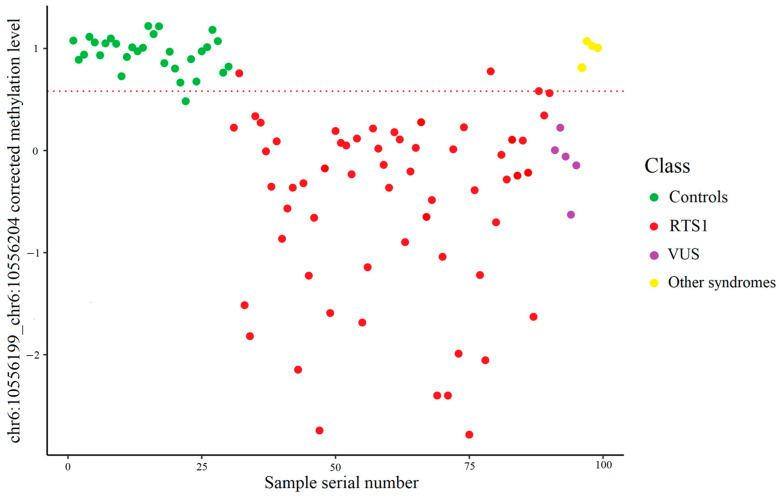
Full dataset samples classification by model based on Sanger bisulfite sequencing. The dashed line represents the cut-off value. All of the samples with VUS in the *CREBBP* gene were classified as RTS1. All of the samples from patients with phenotypically overlapping syndromes clustered above the cut-off line were classified as controls.

**Figure 7 ijms-26-09183-f007:**
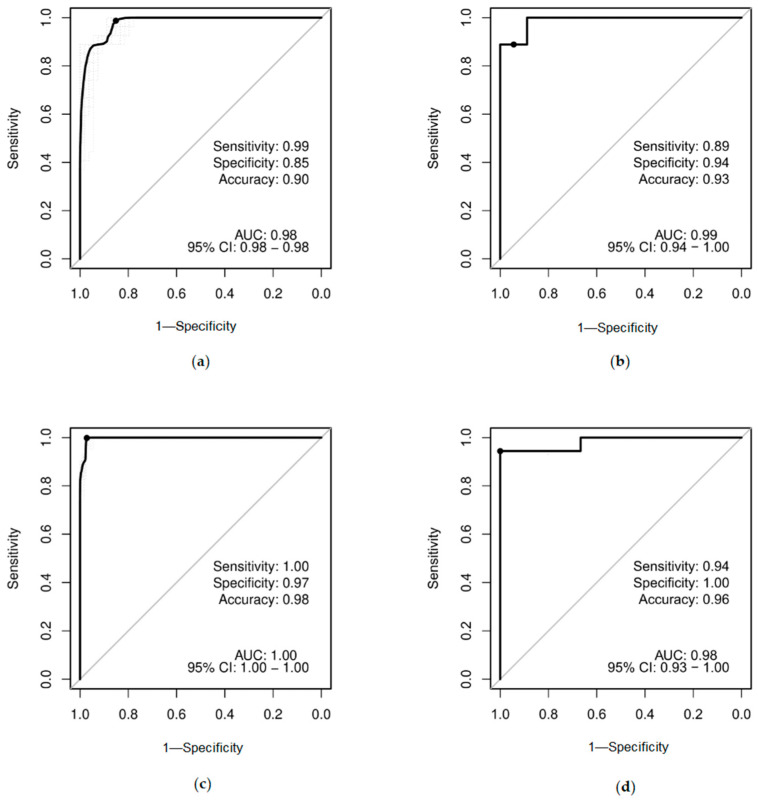
Sensitivity, specificity and accuracy of the classification models presented as ROC curves. (**a**) Training set (model based on targeted high-throughput bisulfite sequencing). (**b**) Test set (model based on targeted high-throughput bisulfite sequencing). (**c**) Training set (model based on Sanger bisulfite sequencing). (**d**) Test set (model based on Sanger bisulfite sequencing).

**Table 1 ijms-26-09183-t001:** CpG dinucleotides selected from the RTS1 episignature for further analysis and corresponding loci; difference in methylation level (Delta Beta) compared to controls [17] and genomic coordinates.

Locus Designation	EPIC BeadChips Probe ID	Delta Beta	Genomic Coordinate (hg19)
1.1	cg12242450	−0.2091863895	chr6:10555808
	cg13347910	−0.1895768459	chr6:10555853
	cg20812045	−0.2185013189	chr6:10555881
1.2	cg12695465	−0.2022041231	chr6:10556147
	cg05869816	−0.2050925327	chr6:10556107
2.1	cg22308949	−0.1705186729	chr2:206628553
	cg04088940	−0.1900411119	chr2:206628454
2.2	cg10807027	−0.2044400041	chr2:206628773
	cg05348875	−0.2226405291	chr2:206628625
	cg10126788	−0.2404856455	chr2:206628727
	cg14157435	−0.2487043485	chr2:206628692
	cg25715429	−0.2529336105	chr2:206628747
	cg20351668	−0.2675453596	chr2:206628737

**Table 2 ijms-26-09183-t002:** Bisulfite PCR primers designed to amplify selected target loci.

Locus Designation	Bisulfite PCR Primers	Insert Coordinates (hg19)	Encompassed CpGs (hg19)
1.1	Primer1F: GGGGGTTAAAGTTTGAATGTTTATTA Primer2R: AAAAAAAATTCCTATCCCAAAAAAT	chr6:10555691-10555946	chr6:10555808 chr6:10555853 chr6:10555881 chr6:10555905
1.2	Primer1F: TTTTTATTTGAGATGTTATTTGTGTTA Primer2R: CCCCCTAATTACTACCATTCAAAA	chr6:10556024-10556275	chr6:10556147 chr6:10556107 chr6:10556052 chr6:10556069 chr6:10556098 chr6:10556174 chr6:10556199 chr6:10556204 chr6:10556244
2.1	Primer1F: TGGGTTTGTTTTTGTTAGATGATAG Primer2R: AACCCACAACAACTTACTCTCCTAA	chr2:206628380-206628588	chr2:206628553 chr2:206628454 chr2:206628415 chr2:206628484 chr2:206628491 chr2:206628521 chr2:206628525 chr2:206628529 chr2:206628531 chr2:206628539
2.2	Primer1F: GGAGAGTAAGTTGTTGTGGGTTATT Primer2R: TAACAAATAAAAATATCATTCAATTACCAT	chr2:206628567-206628809	chr2:206628773 chr2:206628625 chr2:206628727 chr2:206628692 chr2:206628747 chr2:206628737 chr2:206628592 chr2:206628606 chr2:206628609 chr2:206628621 chr2:206628645 chr2:206628714

## Data Availability

The datasets used and/or analyzed during this study are available from the corresponding author upon reasonable request.

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
