# Peer review of "DNA Methylation at a Single Locus of Human Genome Accurately Recapitulates Episignature of CREBBP-Related Rubinstein–Taybi Syndrome [Author-notes fn1-ijms-26-09183]"

_ijms, 2025, doi:10.3390/ijms26189183_

Round 1

Reviewer 1 Report

Comments and Suggestions for Authors

The authors applied targeted quantitative analysis of DNA methylation and found only one locus to be sensitive and accurate in identifying Rubinstein-Taybi Syndrome 1 (RTS1) patients. However, the authors should be cautious about making such a definitive conclusion.

Comments:

Selected CpG locations: Are these CpGs commonly associated with CNVs (Copy Number Variations)?

Line 277: This finding suggests that the methylation levels at chr6:10556199_chr6:10556204 reflect a disruption in the function of the CREBBP gene regardless of phenotype.

 Why is this the case? It may be that the methylation level at chr6:10556199_chr6:10556204 is not unique or sufficiently indicative of RTS1 patients.

Line 365: How do the methylation levels in NGS correlate with those from Sanger sequencing?

Please clarify the comparison between Supp Materials 1, Table S2 and Table S3. In Table S3, several cells under the columns for chr6:10556199 and chr6:10556204 are empty. Please explain this discrepancy.

Line 367: Could you clarify what is meant by “no methyl groups”?

Figure 1:

Please elaborate on the scale of importance shown on the Y-axis.

How did you determine the cutoffs for rejection/confirmation/tentative decisions?

Supp Material 1, Table S1:

VUS: NM_004380.3(CREBBP):c.1941+5G>A is classified as pathogenic in ClinVar.

The authors may reevaluate VUS or pathogenic variants, particularly those that are misclassified by this model, as shown in Figures 5 or 6.

Line 96, Line 329: The study by Yanan T. et al. [12] is cited.

Is this citation correct? The name does not match the reference.

The 12.reference is:
Aref-Eshghi, E.; Kerkhof, J.; Pedro, V.P.; Barat-Houari, M.; Ruiz-Pallares, N.; Andrau, J.-C.; Lacombe, D.; Van-Gils, J.; Fergelot, P.; Dubourg, C.; et al. Evaluation of DNA Methylation Episignatures for Diagnosis and Phenotype Correlations in 42 Mendelian Neurodevelopmental Disorders. Am. J. Hum. Genet. 2020, 106, 356–370.

Author Response

Selected CpG locations: Are these CpGs commonly associated with CNVs (Copy Number Variations)?

Dear Reviewer,

according to gnomAD, selected CpG locations are not commonly associated with CNVs.
There are two extremely rare CNVs, with allele frequencies of 0.00004610 each (i.e., once seen in the whole cohort).

Line 277: This finding suggests that the methylation levels at chr6:10556199_chr6:10556204 reflect a disruption in the function of the CREBBP gene regardless of phenotype.

 Why is this the case? It may be that the methylation level at chr6:10556199_chr6:10556204 is not unique or sufficiently indicative of RTS1 patients.

Yes, we absolutely agree with your comment; it is true that the methylation level of chr6:10556199_chr6:10556204 locus might be insufficiently indicative of RTS1 patients in a certain way, and its uniqueness is to be estimated in the future with the use of a more complete set of samples from phenotypically overlapping patients with other chromatinopathies. By making a statement that “the methylation level at chr6:10556199_chr6:10556204 reflects a disruption in the function of the CREBBP gene regardless of phenotype” we implied that in our dataset only those patients with the CREBBP mutations were classified as “RTS1”, whereas the patients with KMT2A and SRCAP mutations were correctly classified as “non-RTS1” despite the fact that their phenotype was highly suggestive of RTS.

We hope that it is clearly indicated in the same paragraph: “Nevertheless, more confident use of episignature analysis for differential diagnosis requires analyzing a larger number of DNA samples from patients with phenotypically overlapping syndromes”.

Line 365: How do the methylation levels in NGS correlate with those from Sanger sequencing?

Thank you for this excellent question, it is quite a fascinating idea for further research. As far as we noticed, there was no solid tendency in methylation levels obtained by either method (i.e. generally higher or lower values); however, it was not our primary goal to compare the ability of Sanger bisulfite sequencing and targeted high-throughput bisulfite sequencing to measure DNA methylation level. We can certainly say that since there is a slight difference in the methylation levels obtained by these two methods, it is crucial to avoid mixing the data up while assembling the ML-based classification model. We have added these considerations to the Discussion, penultimate paragraph.

Please clarify the comparison between Supp Materials 1, Table S2 and Table S3. In Table S3, several cells under the columns for chr6:10556199 and chr6:10556204 are empty. Please explain this discrepancy.

The discrepancy between the results of Sanger bisulfite sequencing and targeted high-throughput bisulfite sequencing methylation levels measurement was discussed in lines 245-260 (259-266 of the revised manuscript), but your comment helped us to realise that we did not indicate the reason clearly enough. We have added the following part: “However, some loss in DNA methylation level data is observed while using Sanger bisulfite sequencing compared to targeted high-throughput bisulfite sequencing. The lack of data corresponding to certain cytosines in several samples could have occurred due to the imperfection of the context-based signal level measurement, which is inevitable because of the variability of the electropherograms. This also might have been the reason for the discrepancy between the outcome of the feature selection and the predictor that the most accurate classification model was based on.”.

Line 367: Could you clarify what is meant by “no methyl groups”?

By this phrase we meant that all of the methylated cytosines in the model amplicons designed specifically for the methylation level measurement were replaced with nonmethylated cytosines during PCR. We can see how this might seem confusing, so we have changed the wording in both line 367 and line 349. Line 367 (397 of the revised manuscript): “...calculated for each corrected signal in model amplicons with nonmethylated cytosines only”. Line 349 (378 of the revised manuscript): “...model amplicons that contain only nonmethylated cytosines were produced by amplifying target regions of a DNA sample…”.

Figure 1:

Please elaborate on the scale of importance shown on the Y-axis.

How did you determine the cutoffs for rejection/confirmation/tentative decisions?

It is our fault that we failed to provide a reference for the Boruta feature selection algorithm in the “Materials and Methods” section. We have added the following citation: [34] Kursa, M.B.; Rudnicki, W.R. Feature Selection with the Boruta Package. J. Stat. Soft. 2010, 36, 1-13. In this article, it is described in detail how the feature selection with this R package is conducted. In a nutshell, the main idea of this machine learning technique is that it introduces ‘shadow’ attributes, whose values are obtained by shuffling values of the original attributes across objects. The classification is performed by numerous decision trees, and each of them works with certain samples of the training set. The authors claim that “the importance measure of an attribute is obtained as the loss of accuracy of classification caused by the random permutation of attribute values between objects”, and it is computed as a Z score (the average loss of accuracy divided by its standard deviation). To determine which attributes might be considered important, the maximum Z score among shadow attributes (MZSA) is computed; then, the importance of each attribute is compared to this value. Attributes which have importance significantly lower than MZSA are marked as ‘unimportant’, and attributes which have importance significantly higher than MZSA are marked as ‘important’. As for the “tentative” cases, M.B. Kursa and W.R. Rudnicki wrote: “Due to the fact that the number of random forest runs during Boruta is limited by the maxRuns argument, the calculation can be forced to stop prematurely, when there are still attributes which are judged neither to be confirmed nor rejected — and thus finally marked tentative.”

Supp Material 1, Table S1:

VUS: NM_004380.3(CREBBP):c.1941+5G>A is classified as pathogenic in ClinVar.

Yes, this variant is indeed registered in the ClinVar database as pathogenic; however, the lack of evidence does not allow us to trust this source in this particular case. We can only apply PM2 and PP3 criteria at best, and this combination suggests that this variant is of uncertain clinical significance, according to ACMG/AMP guidelines.

The authors may reevaluate VUS or pathogenic variants, particularly those that are misclassified by this model, as shown in Figures 5 or 6.

We agree that it is an extremely tempting idea to shift the pathogenicity of those variants which were misclassified. All of the extensive studies on episignatures indicate that in the near future there might appear a criterion specifically designed to consider the results of the episignature analysis, which would allow geneticists to interpret variants more efficiently. And yet, this area of research is still in its early stage of development; from our point of view, it is somewhat premature to reevaluate variants based on the methylation level data. There are numerous chromatinopathies for which the episignature has not been established yet; therefore, it is still difficult to assess the uniqueness of the methylation patterns of certain loci. Our thoughts on this matter are also reflected in lines 297-323 of the revised manuscript.

Line 96, Line 329: The study by Yanan T. et al. [12] is cited.

Is this citation correct? The name does not match the reference.

The 12.reference is:
Aref-Eshghi, E.; Kerkhof, J.; Pedro, V.P.; Barat-Houari, M.; Ruiz-Pallares, N.; Andrau, J.-C.; Lacombe, D.; Van-Gils, J.; Fergelot, P.; Dubourg, C.; et al. Evaluation of DNA Methylation Episignatures for Diagnosis and Phenotype Correlations in 42 Mendelian Neurodevelopmental Disorders. Am. J. Hum. Genet. 2020, 106, 356–370.

Yes, it is our mistake, thank you for your attentiveness; there should have been indicated [17] instead, which is a study by Yanan Tang et al.

Reviewer 2 Report

Comments and Suggestions for Authors

The core finding of the study is a compelling demonstration of a groundbreaking simplification of episignature diagnostics. The research successfully posits that a single, targeted DNA methylation locus (chr6:10556199_chr6:10556204) can serve as a highly accurate biomarker for Rubinstein-Taybi syndrome type 1 (RTS1). This finding is supported by a robust methodological comparison using both targeted Next-Generation Sequencing (NGS) and a custom quantitative Sanger sequencing method. The study's primary strengths lie in its compelling proof-of-concept for a rapid, cost-effective, and technically accessible diagnostic tool for RTS1. Its ability to correctly classify samples with variants of uncertain significance (VUS) and distinguish RTS1 from phenotypically overlapping syndromes underscores its significant clinical utility. However, the research also presents several key limitations that merit careful consideration. A central discrepancy exists between the two analytical methods: the Sanger-based model, despite significant data loss, paradoxically showed slightly higher performance metrics than the more comprehensive NGS-based model. Furthermore, the sample sizes for the critical validation cohorts—those with VUS (n=5) and those with phenotypically overlapping syndromes (n=4)—are notably small. This limits the statistical power and generalizability of these crucial findings.

A notable strength of the study is its side-by-side comparison of two distinct quantitative DNA methylation analysis methods: targeted high-throughput bisulfite sequencing (NGS) and a custom quantitative Sanger bisulfite sequencing method. This comparison is a valuable contribution to the field, as it provides a practical technical validation of different platforms for a specific clinical application. 

The most significant methodological limitation of the study is the unresolved discrepancy between the performance of the Sanger and NGS-based models. The Sanger-based model, despite the documented loss of data for several CpG sites, surprisingly demonstrated superior performance on the test set, with a specificity of 1.00. The authors attribute this to the algorithm's decision-making based on incomplete data, but this explanation is insufficient.

Please include a graphical abstract for a better illustration of the study design and results obtained. 

Please write a separate limitation section and mention the limitations of the study.

Author Response

The most significant methodological limitation of the study is the unresolved discrepancy between the performance of the Sanger and NGS-based models. The Sanger-based model, despite the documented loss of data for several CpG sites, surprisingly demonstrated superior performance on the test set, with a specificity of 1.00. The authors attribute this to the algorithm's decision-making based on incomplete data, but this explanation is insufficient.

Thank you for this commentary. We tend to explain this discrepancy by the limited number of samples in our cohort, which, in the process of random splitting into sets, might slightly affect the resulting numbers. Anyway, the numbers for both methods are superior, and we consider the discrepancy not critically significant. We expect that more exact numbers will be achieved on greater cohorts.

Please include a graphical abstract for a better illustration of the study design and results obtained.

Please write a separate limitation section and mention the limitations of the study.

It is our greatest pleasure to receive such detailed feedback. We genuinely appreciate your expert comments which helped us to enhance the quality of our work. As you suggested, we have added the following paragraph mentioning the limitations of our study in the “Discussion” section (lines 323-335 of the revised manuscript):

“Overall, despite the fact that both models showed remarkable results in classifying samples from healthy individuals and patients suffering from RTS1, Wiedemann-Steiner syndrome, Floating-Harbor syndrome and Cornelia de Lange syndrome, it is crucial to note the limitations of the current study. The existing state of the Sanger bisulfite sequencing methodology does not allow to measure the DNA methylation level as thoroughly as in the case of high-throughput bisulfite sequencing, which makes it difficult to draw any conclusions regarding the comparison between the performances of the models. It should be noted that there is a slight difference in the methylation levels obtained by these two methods in our study, thus it is crucial to avoid mixing the data up while assembling the machine learning based classification model. Furthermore, it is necessary to test the reduced RTS1 episignature on larger cohorts of patients with VUS in the CREBBP gene and patients suffering from phenotypically overlapping disorders in order to better understand its diagnostic value and discriminating accuracy.”

We have also taken your suggestion about including graphical abstract into consideration, and have attached it with the resubmission.

Reviewer 3 Report

Comments and Suggestions for Authors

The study by Zemlianaia et al., provides an episignature with good discriminating power in detecting patients with Rubinstein-Taybi syndrome. Overall, the paper is interesting and well-written. I have prepared a few suggestions that may help the authors in providing a revised version of their work for further consideration.

  1. It could be valuable to provide information regarding the broader context in which the differentially methylated sites reside and especially for the most predictive ones. Do they fall into enhancers? Publicly available data from large consortia could help to provide snapshots with enhancer markers. Also these sequences are bound by CBP-p300 at least in tissues from healthy individuals? It would be interesting for future experiments to examine the chromatin state and the transcription factors that bind to these sites in patients vs controls.
  2. In the discussion, please compare your results with other methylation studies of the particular or analogous syndromes.
  3. I suggest that the authors upload their raw sequencing data on a suitable repository like GEO, so that interested readers can easily assess their results. Moreover, the sanger sequencing files (at least some representatives) could be uploaded as supplementary material with the paper.
  4. In line 97 the authors state that the difference in methylation levels should equal or exceed 20%, but in table, several beta values below 0.2 are presented. Does the combined delta beta of each locus exceeds 20% in all cases? Please clarify and correct if appropriate.
  5. In line 137 does 8000 refers to reads? It could also be written as 8000X.
  6. Please state more explicitly that the number of samples with syndromes phenotypically overlapping with Rubinstein-Taybi syndrome is low to derive strong conclusions about the discriminating accuracy of the episignature.
  7. Please provide references for all bioinformatics tools used in the study.
  8. Please correct a typo (mediane) in Supplementary material 2

Author Response

  1. It could be valuable to provide information regarding the broader context in which the differentially methylated sites reside and especially for the most predictive ones. Do they fall into enhancers? Publicly available data from large consortia could help to provide snapshots with enhancer markers. Also these sequences are bound by CBP-p300 at least in tissues from healthy individuals? It would be interesting for future experiments to examine the chromatin state and the transcription factors that bind to these sites in patients vs controls.

Thank you for this excellent suggestion. As soon as we discovered that the methylation level of chr6:10556199_chr6:10556204 is quite a successful predictor, we found out that this locus is located in the GCNT2 gene promoter region. To the moment, this gene had not been implicated in the RTS1 pathogenesis, but p300 is indeed recruited to the promoter region of this gene in neural cells (although the cluster score is relatively low), according to the ENCODE database. We agree that this finding is an intriguing idea for further research; thus, we have added the following paragraph in the “Discussion” section:

“The data obtained in the current study also provides basis for further research concerning the functional significance of the aberrantly methylated chr6:10556199_chr6:10556204 locus in RTS1 patients. This site is located in the promoter region of the GCNT2 gene, which encodes the enzyme responsible for formation of the blood group I antigen. It could be valuable to investigate if hypomethylation of the chr6:10556199_chr6:10556204 locus affects the expression of GCNT2. It is not clear yet, whether the abnormal level of expression of this gene could contribute to the RTS1 phenotype or not; however, the existing data from the ChIP-seq database ENCODE suggests that p300 is recruited to the area in which the studied CpGs are encompassed, particularly in neural cells.”

In the discussion, please compare your results with other methylation studies of the particular or analogous syndromes.

Considering that our study is the first one to resort to targeted methods of DNA methylation analysis while assessing episignature, it is rather challenging to conduct a comprehensive comparison between our work and those which were based on genome-wide sequencing approaches. But we agree that it is essential to mention previous studies and their contribution to the development of the RTS1 episignature. We have added the following part in the “Discussion” section:

“In the current study we demonstrate that the analysis of the methylation level at only one locus of the genome can classify samples as RTS1 or non-RTS1 with high accuracy. The RTS1 genome-wide episignature was successfully derived in several recent studies [12, 13, 17], and it was shown to be highly specific. We demonstrate how the reduction of the number of studied CpGs from hundreds to just two of them does not result in the loss of accuracy. Both targeted high-throughput bisulfite sequencing and Sanger bisulfite sequencing showed impressive results in capturing minor differences in DNA methylation level between control and RTS1 samples.”

  1. I suggest that the authors upload their raw sequencing data on a suitable repository like GEO, so that interested readers can easily assess their results. Moreover, the sanger sequencing files (at least some representatives) could be uploaded as supplementary material with the paper.

With the Supplementary material we now provide screenshots of both Sanger and NGS for representative control and RTS1 samples.

  1. In line 97 the authors state that the difference in methylation levels should equal or exceed 20%, but in table, several beta values below 0.2 are presented. Does the combined delta beta of each locus exceeds 20% in all cases? Please clarify and correct if appropriate.

The 20% cut-off refers to the mean methylation level of the locus (that is, the mean methylation level of locus 1 which consists of loci 1.1 and 1.2, and the mean methylation level of locus 2 which consists of loci 2.1 and 2.2). We can see how this appears confusing, so we have changed the wording:

“...the difference in methylation level of the selected locus between RTS1 and control samples should equal or exceed 20%”

  1. In line 137 does 8000 refers to reads? It could also be written as 8000X.

Yes, it refers to the number of reads. Thank you, we have corrected this mistake.

  1. Please state more explicitly that the number of samples with syndromes phenotypically overlapping with Rubinstein-Taybi syndrome is low to derive strong conclusions about the discriminating accuracy of the episignature.

We absolutely agree with your comment. We have added a paragraph which describes the limitations of our study, including sample size (lines 323-335):

“Overall, despite the fact that both models showed remarkable results in classifying samples from healthy individuals and patients with RTS1, Wiedemann-Steiner syndrome, Floating-Harbor syndrome and Cornelia de Lange syndrome, it is crucial to note the limitations of the current study. The existing state of the Sanger bisulfite sequencing methodology does not allow to measure the DNA methylation level as thoroughly as in the case of high-throughput bisulfite sequencing, which makes it difficult to draw any conclusions regarding the comparison between the performances of the models. Furthermore, it is necessary to test the reduced RTS1 episignature on larger cohorts of patients with VUS in the CREBBP gene and patients with phenotypically overlapping disorders in order to better understand its diagnostic value and discriminating accuracy.”

  1. Please provide references for all bioinformatics tools used in the study.

Thank you for this useful remark. We have added the following missing references:

  1. Kursa, M.B.; Rudnicki, W.R. Feature Selection with the Boruta Package. J. Stat. Soft. 2010, 36, 1-13.
  2. Kuhn, M. Building Predictive Models in R Using the Caret Package. J. Stat. Soft. 2008, 28, 1-26.
  1. Please correct a typo (mediane) in Supplementary material 2

Corrected.

Round 2

Reviewer 3 Report

Comments and Suggestions for Authors

The authors have addressed my comments adequately and the quality of the paper has been improved.